# LC-MS Metabolite Profiling and the Hypoglycemic Activity of *Morus alba* L. Extracts

**DOI:** 10.3390/molecules27175360

**Published:** 2022-08-23

**Authors:** Qing Yi-Jun Zhou, Xin Liao, Hao-Ming Kuang, Jia-Yu Li, Shui-Han Zhang

**Affiliations:** 1Science and Technology Innovation Center, Hunan University of Chinese Medicine, Changsha 410208, China; 2Institute of Chinese Materia Medica, Hunan Academy of Chinese Medicine, Changsha 410013, China

**Keywords:** *Morus alba* L., mulberry, LC-MS, metabolite profiling, hypoglycemic activity, fragmentation pathways

## Abstract

*Morus alba* L. is used in traditional Chinese medicine for its anti-diabetic activity; however, the part of the hypoglycemic activity and related active metabolites are still not fully clarified. In this study, the metabolites in the *M. alba* roots, leaves, twigs, and fruits extracts (70% ethanol extracts) were systematically identified, and their hypoglycemic activity was evaluated by the high-fat diet/streptozotocin-induced 2 diabetes mellitus (T2D) mouse model. A total of 60 high-level compounds, including 16 polyphenols, 43 flavonoids, and one quinic acid, were identified by high-performance liquid chromatography/quadrupole time-of-flight mass spectrometry (HPLC-Q-TOF-MS) combined with the fragmentation pathways of standards and the self-established database. Among them, 23 metabolites were reported for the first time from this plant. In contrast to the extracts of *M. alba* leaves and fruits, the extracts of roots and twigs displayed significant hypoglycemic activity The glycemia was significantly reduced from 32.08 ± 1.27 to 20.88 ± 1.82 mmol/L and from 33.32 ± 1.98 to 24.74 ± 1.02 mmol/L, respectively, after 4 weeks of treatment with roots and twigs extracts. Compound **46** (morusin), which is a high-level component identified from the extracts of *M. alba* roots, also displayed significant activity in decreasing the blood glucose level of T2D mice reduced from 31.45 ± 1.23 to 23.45 ± 2.13 mmol/L. In addition, the extracts of roots and twigs displayed significant activity in reducing postprandial glycemia. This work marks the first comparison of the metabolites and hypoglycemic activity of *M. alba* roots, leaves, twigs, and fruits extracts, and provides a foundation for further development of *M. alba* extracts as anti-diabetic drugs.

## 1. Introduction

*Morus alba* L. (mulberry), which belongs to the genus of *Morus* from the Moraceae family, is widely distributed in east Asian countries, such as China, Korea and Japan [1,2]. All parts of this plant, including roots, leaves, twigs and fruits, have been used as traditional Chinese medicines for thousands of years and are recorded in the medicinal book *Compendium of Materia Medica*, which is a famous Chinese encyclopedia of medicine written by Shizhen Li in the Ming dynasty [3,4]. The pharmacological activities of *M. alba*, having anti-diabetic, anti-oxidative, anti-bacterial, anti-inflammatory, anti-atherogenic, and immunological enhancement properties, have been verified by the modern experimental technologies [5,6,7]. Previous phytochemistry studies indicated that *M. alba* mainly contains flavonoids, polyphenol, anthocyanins, terpenes, carotenoids, and alkaloid-type metabolites, such as kuwanon G, morusin, and 1-deoxynojirimycin [8,9,10]. In addition to these well-known compounds, many unknown metabolites still need to be investigated.

Chemical constituents undergo continual changes during plant growth, and metabolites vary in the roots, leaves, twigs and fruits of *M. alba* [11,12]. In some previous studies [13,14,15], liquid chromatography-mass spectrometry (LC-MS) were employed to identify the compounds of *M. alba* samples that were taken only at a single part of the plant, which caused a limited number of compounds to be detected and identified. In addition, the differences in chemical constituents from different parts of *M. alba* have rarely been investigated. Therefore, in the present study, we comprehensively analyzed compounds, which may contribute to hypoglycemic activity, in the roots, leaves, twigs and fruits of this plant by HPLC-Q-TOF-MS.

Diabetes is one of the most prevalent metabolism-related diseases that significantly affects human health all over the world [16,17]. According to the International Diabetes Federation (IDF, 2019), diabetes is predicted to affect 578 million in 2030 [18,19], and T2D accounts for approximately 90% of total diabetic patients. Synthetic medications for T2D lead to many adverse effects coupled with expensive costs. Herbal medicines and natural ingredients have investigated in many countries and regions due to their significant anti-diabetic activity, low adverse effects, and low costs [20,21]. Therefore, developing new anti-diabetic agents from traditional herbal medicines has been a research focus. The anti-diabetic activity of roots, twigs or leaves extracts of *M. alba* have been reported in previous studies [22,23,24]; however, comparing the hypoglycemic activity of roots, leaves, twigs and fruits extracts and analysing the corresponding active components simultaneously has been done rarely. Therefore, this study evaluated the hypoglycemic activity of four parts of *M. alba* using a T2D mouse model and identified the main metabolites by LC-MS technology.

In summary, the primary metabolites of the extracts of *M. alba* roots, leaves, twigs and fruits were primarily screened and identified by HPLC-Q-TOF-MS, and then the hypoglycemic activity of four extracts, and one high-level compound (**46**), identified from root extracts were evaluated by a T2D mouse model.

## 2. Results and Discussion

### 2.1. Investigation of the Fragmentation Pathways of Standards

A series of metabolites with the same skeleton but various substituent groups are present in different parts of *M. alba*. Since these analogues typically display similar fragmentation patterns in their MS/MS spectra, investigating the fragmentation behaviors of well-characterized standards is a valid strategy for identifying unknown analogue structures. In this study, a total of six references, including chlorogenic acid (**5**), rutin (**11**), isochlorogenic acid A (**15**), kuwanon G (**31**), kuwanon H (**36**), and morusin (**46**), which were reported from *M. alba* in previous studies [25], were employed for investigating the fragmentation pathways of polyphenols, flavonoids and flavonoid glycosides distributed in this plant.

In the MS/MS spectra of compounds **5** and **15** (Figure 1A,C), the primary fragmentation pathway was the cleavage of the bond between quinic acid and caffeic acid, and the formation of the high abundance ions at *m*/*z* 353.0867 and 191.0558 (or 191.0557 for quinic acid) was seen [11]. Three fragmentation pathways predominated in the MS/MS spectra of compounds **11**, **31**, **36** and **46** (Figure 1B,D–F). The first fragmentation was a neutral loss of all sugar moieties or substituent groups, which was connected to the flavonoid skeleton, from the deprotonated flavonoid and formation of the high abundance fragments. In the MS/MS spectra of compounds **11**, **31**, and **36**, the high abundance ions at *m*/*z* 300.0274, 353.1043, and 353.1004 were produced by the loss of all sugar moieties or substituent groups from the deprotonated flavonoids at *m*/*z* 609.1458, 691.2108, and 759.2802, respectively (Figure 1B,D,E). The second fragmentation behavior was the cleavage of the C-ring of the flavonoid skeleton and the formation of a series of fragments in the low *m*/*z* zone, which was investigated in previous studies [11,12]. The third fragmentation pattern was the loss of some small substituent groups, such as H_2_O, CO, and the catechol moiety, from the parent ion. Take compounds **31**, **36**, and **46** for example. The fragment ions at *m*/*z* 109.0307, 109.0291, and 109.0289 were formed by the loss of a catechol moiety from the parent ions at *m*/*z* 691.2108, 759.2802, and 419.1503, respectively. The loss of catechol moiety from the parent ions and the formation of the fragments at *m*/*z* 109.0290 (theoretical value) was rarely reported in previous studies.

### 2.2. Identification of Polyphenols and Flavonoids from M. alba

Total ion chromatograms (TICs) of *M. alba* roots, leaves, twigs, and fruits (Figure 2) were primarily produced using the HPLC-Q-TOF-MS. The results showed that the roots contain more high-level metabolites than other parts of *M. alba*. The MS/MS spectra of metabolites, which display obvious peaks in the TICs, were obtained by the target MS/MS method. Finally, a total of 60 metabolites (Table 1), which may contribute to hypoglycemic activity, were screened and identified by their MS/MS spectra combined with the well-investigated fragmentation pathways of standards and the self-established database (Appendix A). In addition, the structures of high-level metabolites **5**, **11**, **15**, **31**, **36** and **46** were unambiguously determined by comparing the retention time, MS, and MS/MS data with the references (Figure 3). 

A total of 16 metabolites including **2**, **3**, **5**, **6**, **8**, **15**, **16**, **18**, **21**, **22**, **24**, **32**, **33**, **41**, **45**, and **52** were identified as polyphenols based on their MS/MS spectra (Appendix A). Take compound **16** as an example. The MS fragmentation pathways of compounds **5** and **16** were highly similar, and the difference *m*/*z* value between both compounds was 19.9945 Da, which indicated that one of the hydroxyl groups was replaced by a hydrogen atom and formed the structure of metabolite **16**. In the MS/MS spectrum of compound **16** (Figure 4A), the fragment ions at *m*/*z* 163.0380 demonstrated that the hydroxyl group in the caffeic acid part was replaced. Therefore, compound **16** was tentatively identified as 5-O-p-coumaroylquinic acid, the fragmentation pathways having been reported in a previous study [11] and was reported for the first time from this plant. The high abundance peak **2** gives a mass-to-charge ratio at *m*/*z* 567.1722, which was used for screening its possible structure in the self-established database (Appendix A) that collected all reported compounds from the genus of Morus. The candidate named mulberroside A (theoretical *m*/*z* 567.1714, [M − H]^−^) was screened. In the MS/MS spectrum of compound **2** (Figure 4B), the continual loss of a glucose moiety from the deprotonated parent ions at *m*/*z* 567.1722, and formation of a high abundance of fragment ions at *m*/*z* 405.1197 and 243.0660, were observed. The fragmentation pathways of compound **2** are in accordance with the structure of mulberroside A; therefore, compound **2** was tentatively identified as mulberroside A. Using the above methods, the remaining 13 polyphenols (**3**, **6**, **8**, **15**, **18**, **21**, **22**, **24**, **32**, **33**, **41**, **45**, and **52**) were identified (Figure 3 and Appendix A, and Table 1).

A total of 43 metabolites, including **4**, **7**, **9**–**14**, **17**, **19**, **20**, **23**, **25**–**31**, **34**–**40**, **42**–**44**, **46**–**52**, and **54**–**60**, were identified as flavonoids or flavonoid glucosides based on their MS/MS spectra (Appendix A). Take compound **10**, **19**, and **35** structural identifications for examples. In the MS/MS spectrum of compound **10** (Figure 4C), the loss of a glucoside moiety from the parent ion at *m*/*z* 463.0879 and the formation of the characteristic basic fragments at *m*/*z* 300.0276 (the ion was also appeared in the MS/MS spectrum of standard **11** and was regarded as the characteristic ion of quercetin) were observed. Therefore, compound **10** was tentatively identified as quercetin-3-O-glucopyranoside, which has never been reported from this plant. Peak **19** gives a mass-to-charge ratio at *m*/*z* 625.1707, which was used for screening its possible structure in Appendix A. The only candidate, named kuwanon J with theoretical values of *m*/*z* 567.1714 ([M − H]^−^), was screened (Appendix A). In the MS/MS spectrum of compound **19** (Figure 4D), the high abundance of ions at *m*/*z* 499.1390 and 125.0239 was formed due to the cleavage of the C-ring of the parent ion at *m*/*z* 625.1707. The loss of catechol moiety from the precursor ion at *m*/*z* 499.1390 and the formation of the fragments at *m*/*z* 389.1024 were observed. The above fragmentation behaviors of compound **19**, which were similar to those of standards **31** and **36**, were matched with the candidate’s structure. Therefore, compound **19** was identified as Kuwanon J, which was reported for the first time from *M. alba*. The fragmentation pathways of compounds **35** and **46** (standard) were highly similar, and the difference *m*/*z* value between both compounds was 2.0150 Da, which indicated that one of the double bonds was oxidized or the oxygen-ring connected to the A-ring was opened and formed the structure of metabolite **35**. In the MS/MS spectrum of compound **35** (Figure 4E), the fragment ions at *m*/*z* 193.0867 demonstrated that the oxygen-ring connected to A-ring was opened. Therefore, compound **35** was identified as mulberrin, which was reported from this plant in a previous study. Using above methods, the remaining 36 flavonoids (**4**, **7**, **9**, **12**–**14**, **17**, **20**, **23**, **25**–**30**, **34**, **37**–**40**, **42**–**44**, **47**–**52**, and **54**–**60**) were identified (Figure 3 and Appendix A, and Table 1).

Finally, a total of 60 metabolites, including 43 flavonoids, 16 polyphenols, and one quinic acid, were identified using their MS/MS spectra, fragmentation pathways of standards and a well-established database, among which, 23 compounds (**1**, **3**, **7**–**10**, **12**, **14**–**16**, **19**, **23**, **25**, **26**, **28**, **30**, **34**, **37**, **39**, **44**, **47**, **48**, and **57**) were reported for the first time in this plant (Table 1). The primary metabolites of *M. alba* roots, leaves, twigs, and fruits, such as compounds **31**, **35**, **36**, **40**, **46**, **52** and **59**, were in this study, and may contribute to the hypoglycemic activity of the extracts.

The distribution of all identified metabolites in the roots, leaves, twigs, and fruits of *M. alba* was determined using extracted ion chromatography (EIC) based on the TICs. All identified compounds were found in the *M. alba* roots, and the species and number of metabolites in the roots were more abundant than in other parts. Polyphenols and flavonoids were the primary metabolites of roots and twigs. Most of the high-level prenylated flavonoids, such as compounds **31**, **34**, **35**, **36**, **40**, **46**, **50**, and **52**, were only detected from *M. alba* roots and twigs; however, it was difficult to find those compounds from the leaves and twigs. In addition, the number of identified polyphenols in the leaves was more than that of fruits. In summary, the identified metabolites in the four parts of *M. alba* were different (Table 1), which explains why the roots, leaves, twigs, and fruits were used as different medicines in traditional Chinese medicine.

### 2.3. Evaluation of the Hypoglycemic Activity for Different Extracts of M. alba

To evaluate the hypoglycemic activity of roots, leaves, twigs, and fruits, 70% ethanol extracts of four parts (namely RM, LM, TM, and FM) and a high-content compound named morusin (**46**) were employed. The result showed that the blood glucose level of the model control group (MC) was significantly higher (*p* < 0.01) compared with the normal control group (NC) at 0, 7, 14, 21, and 28-days, respectively, which indicated that a high blood glucose mouse model (T2D) was successfully established (Figure 5A, Appendix A). The blood glucose level of the acarbose group at 7 (24.12 ± 0.79 mmol/L), 14 (24.56 ± 0.43 mmol/L), 21 (23.78 ± 0.78 mmol/L), and 28-days (22.21 ± 0.73 mmol/L) significantly decreased (*p* < 0.05) compared with the initial level at 0-days (33.08 ± 0.56 mmol/L), which indicated that the positive group displays significant hypoglycemic activity. Compared with the initial blood glucose level at 0 days (32.08 ± 1.27 and 33.32 ± 1.98 mmol/L for RM and TM, respectively), the RM (21 (22.21 ± 2.03 mmol/L) and 28-days (20.88 ± 1.82 mmol/L)) and TM (28 days (24.74 ± 1.02 mmol/L)) groups were significantly decreased (*p* < 0.05 or *p* < 0.01), which indicated that the daily administration of the roots and twigs extracts have remarkable hypoglycemic activity, especially for the extract of roots (Figure 5A, Appendix A). The blood glucose level of the morusin (**46**) group (MS) at 28 days (23.45 ± 2.13 mmol/L) was significantly decreased (*p* < 0.05) compared with the initial level at 0-days (31.45 ± 1.23 mmol/L), which indicated that this compound has noteworthy hypoglycemic activity. However, the daily administration of leaves and fruits extracts did not display significant activity in decreasing blood glucose levels.

As shown in Figure 5B (Appendix A), the body weight of mice in the normal group was significantly increased from 0 to 28 days (*p* < 0.05), however, the body weight of mice in the control group was significantly decreased (*p* < 0.05), which was in accordance with the body weight being decreased in diabetic patients. Compared with the normal control group, the body weight of the MC group at 21 and 28 days was significantly decreased (*p* < 0.01). The body weight of the RM group at 28 days was significantly increased (*p* < 0.05) compared with the initial level at 0 days, which indicated that the RM group displayed significant ability to reverse the T2D-associated weight loss in diabetic mice. However, the acarbose, morusin, and the extracts of leaves, twigs, and fruits did not have that functions.

The above results indicate that the extracts of *M. alba* roots and twigs have excellent activity for decreasing the blood glucose level in T2D disease. Morusin (**46**), which was one of the high-level components of the extracts of *M. alba* roots, also showed significant hypoglycemic activity. However, the hypoglycemic activity of compound **46** was weaker than that of roots extract, which indicates the presence of other compounds with hypoglycemic activity in the root extract. The extracts of leaves and fruits did not display significant activity in decreasing blood glucose levels, which could suggest that the high-level prenylated flavonoids, such as compounds **31**, **36**, **40**, **46**, and **52**, presented in the extract of *M. alba* roots might contribute to its hypoglycemic activity. Metabolite **52**, which was rich in the extracts of roots and twigs, and absent in that of leaves and fruits, was further regarded as a hypoglycemic ingredient by a comprehensive analysis of the level of this compound in the four extracts and their hypoglycemic activities. In addition, the structure of compound **52** (Morusinol) was very similar to that of metabolite **46**, which indicated that the high-level compound **52** has potential hypoglycemic activity and requires further study. 

### 2.4. The Effect of Roots and Twigs Extracts on the Postprandial Glycemia

To explore the potential mechanism of hypoglycemic activity, a postprandial glycemia experiment for the roots and twigs extracts was carried out. The results show that the acarbose (10.34 ± 0.23 mmol/L), roots (9.01 ± 0.47 mmol/L) and twig extract (10.25 ± 1.11 mmol/L) groups displayed significant activity in reducing postprandial glycemia (*p* < 0.05) compared to the normal control group (13.21 ± 1.11 mmol/L) after administration of corresponding solutions for 30 min (Figure 6, Appendix A). The extracts of *M. alba* roots displayed stronger activity than the positive control (AB) and the twig extracts (TM) for reducing postprandial glycemia. The blood glucose level of all groups returns to baseline after 120 min.

Four main mechanisms of hypoglycemic activity for the *M. alba* roots, twigs, and leaf extracts were investigated in previous studies. The first one involved inhibiting the intestinal α-glucosidase activity, thereby blocking the supply of glucose [26,27]. The second involved improving glucose uptake and promoting glycogen synthesis by activating the corresponding signaling pathway or protein [6,28]. The third involved inhibiting oxidative stress, apoptosis, and inflammation and protecting the pancreatic islet cells [29,30]. The last one involved decreasing the level of total cholesterol, triglyceride, and low-density lipoprotein cholesterol and increasing the high-density lipoprotein cholesterol further to reduce the blood glucose content [31]. In this study, the extracts of *M. alba* roots and twigs displayed significant hypoglycemic activity and reduced postprandial glycemia function, which agreed with a previous report [32]. The postprandial glycemia experiment results indicated that inhibiting the intestinal α-glucosidase activity to block the supply of glucose was the potential mechanism of hypoglycemic activity. In addition, the metabolites **11** (Rutin), **31** (Kuwanon G) and **46** (Morusin) identified from the extracts of *M. alba* roots and twigs were α-glucosidase inhibitors, according to previous reports [26,28]. Therefore, inhibiting the intestinal α-glucosidase activity, and thereby reducing postprandial glycemia, could be a primary mechanism of hypoglycemic activity for the *M. alba* roots and twigs, especially for the extract of roots.

## 3. Materials and Methods 

### 3.1. Materials and Chemicals

Methanol (AR) was purchased from National Institutes for Food and Drug Control (Beijing, China), which was used for extraction. Acetonitrile (HPLC-grade) was obtained from Merck (Darmstadt, Germany), and deionized water was purified using a Milli-Q system (MA, USA). Acarbose (purity ≥ 95%), sucrose, and streptozotocin (STZ) was purchased from National Institutes for Food and Drug Control (Beijing, China). Six references, including chlorogenic acid (**5**), rutin (**11**), isochlorogenic acid A (**15**), kuwanon G (**31**), kuwanon H (**36**), and morusin (**46**), were obtained from Chengdu Must Bio-Technology Co., Ltd. (Sichuan, China).

### 3.2. Extracts Preparation 

The roots, leaves, twigs, and fruits were collected in June 2021 from five-year-old *M. alba* (Xiangsang-6, *n* = 3) in the Hunan Agricultural University (Hunan, China, coordinates: 113°04′25.55″ E, 28°10′45.11″ N). They were authenticated by Prof. Li Chen (Hunan Agricultural University, China). The fresh roots, leaves, twigs, and fruits were dried in a vacuum drying oven at 50 °C for 72 h, and most of the dried samples were crushed by a disintegrator. The rest of the samples were stored at Hunan University of Chinese Medicine (stored number: *M. alba*-20200625). Approximately 100.0 g of sample powder was suspended in 1000 mL 70% ethanol-water (*v*/*v*). Extraction was carried out in an ultrasonic bath for 120 min, and the extract solution was filtered through a 0.22 μm nylon membrane. Four dried extracts were obtained after recovering the solvent by vacuum reduction concentration. 

### 3.3. HPLC-Q-TOF-MS Conditions

Chromatography was performed using an Agilent 1290 HPLC system (Agilent Technologies, Santa Clara, CA, USA). A XAqua C18 (150 mm × 2.1 mm, 2.8 µm; Accrom Technologies Co., Ltd., DaLian, China) was used as a separation column. The elution solution consisted of deionized water (A) and acetonitrile (B). The elution program was as follows: 0–10 min, 5–38% B; 10–25 min 38–60% B; 25–40 min 60–90% B. The rate was set at 0.35 mL/min, and the injection volume was 2 μL. 

Mass spectrometric experiments were performed using a 6530 Q-TOF/MS accurate mass spectrometer (Agilent Technologies) in negative mode. The condition of Q-TOF-MS was optimized as follows: gas temperature and sheath gas temperature were both set at 350 °C, sheath gas and drying gas flow were both optimized to 11 L/min, and fragmenter voltage was set at 150 V. The Q-TOF-MS was continuously calibrated using a reference solution at *m*/*z* 112.9855 and 966.0007 to obtain the precise mass data. The MS/MS data of each metabolite were obtained using the target-MS/MS mode.

### 3.4. Experimental Animal

Institute of Cancer Research (ICR) mice (4 weeks old, male, weight range of 18.0–22.0 g) were purchased from Hunan Slake Jing-da Experimental Animals Co., Ltd. (Certificate number 43004700048590). Animals were housed under a standard 12: 12 h light/dark schedule, and the ambient temperature was controlled at 22 ± 2 °C. All experiments and procedures were carried out according to the Regulations of Experimental Animal Administration issued by the State Committee of Science and Technology of China.

### 3.5. Establishment of T2D Mouse Model and Drug Administration

In addition to 10 mice in the normal control group (NC, providing a normal diet), the other 80 mice were subjected to a high-fat diet for 30 days, and 70 mice (Blood glucose level > 11.6 mmol/L) [33] were successfully induced to the T2D mouse model by intraperitoneal injection of STZ for three days (80 mg/kg) after the high-fat diet feeding. The diabetic mice (68 days old) were randomly divided into seven groups, each containing 10 mice. The normal and model control groups were given distilled water by intragastric administration. The other groups were given the corresponding solution of extracts or drugs (distilled water was used as the solvent; 200 mg/kg for the extracts, and 50 mg/kg for acarbose and morusin (**46**), respectively) with a volume of 20 mL/kg (Appendix A). Mice were treated daily with the corresponding solutions and a high-fat diet for the following 28 days. Blood from the tail was collected at 9:00 am after 12 h of diet prohibition on the 0, 7, 14, 21, and 28 days. The blood glucose level and body weight were determined by a glucometer (Johnson, Elkhart, IN, USA) and laboratory balance, respectively, every 7 days for 28 days.

### 3.6. Postprandial Glycemia of the Normal Mice

Twenty-four normal mice (4 weeks old, weight range of 18.0–22.0 g) were randomly divided into four groups based on their weight, each containing six mice. All mice were given sucrose solution with 2.0 g/kg by intragastric administration after 12 h of diet prohibition. In addition to the normal control group (NC, providing distilled water), the other three groups were given acarbose (50 mg/kg), root and twig extract solutions (200 mg/kg for RM and TM), respectively, after the supply of sucrose solution, using the same mean of administration. Blood from the tail was collected after intragastric administration of extract solution, and that time was defined as zero time (0 min). The blood glucose level was further determined by a glucometer at 0, 30, 60, 90, and 120 min.

### 3.7. Statistical Method

SPSS 16.0 statistics software (Inc., Chicago, IL, USA) was used for statistical analysis; the statistically significant level was set to *p* ≤ 0.05. The data were expressed as mean ± standard deviation. Statistical differences and biological significance were considered simultaneously in the evaluation. 

## 4. Conclusions

In this study, the fragmentation pathways of six standards were investigated in detail. A total of 60 high-level metabolites including 16 polyphenols, 43 flavonoids, and one quinic acid were screened and identified from *M. alba* root, leaf, twigs and fruit extracts using HPLC-Q-TOF-MS, combined study of fragmentation behavior of references using a self-established database. Twenty-three compounds were reported from this plant for the first time. The types and amounts of metabolites in the root extract were greater than that of other parts of *M. alba*. Finally, the hypoglycemic activity of *M. alba* root, leaf, twig, and fruit extracts were evaluated using a T2D mouse model. The results show that the *M. alba* root, and twig extracts displayed significant hypoglycemic activity, but the extracts of leaves and fruits did not show noteworthy hypoglycemic activity. In addition, the hypoglycemic activity of compound **46**, named morusin, which was one of high-level metabolite identified from *M. alba* roots extract, was assessed. The results showed that morusin resulted in a significant decrease of blood glucose level, which indicated that this metabolite is an active component contributing to the hypoglycemic activity of the *M. alba* root extract. The chemical components and hypoglycemic activity of *M. alba* root, leaf, twig, and fruit extracts were investigated simultaneously in this study, and we considered high-level prenylated flavonoids, such as compounds **31**, **36**, **40**, **46**, and **52**, as potential hypoglycemic metabolites. In addition, metabolites **36** and **52**, which were rich in the extracts of roots and twigs, and absent in that of leaves and fruits, were further regarded as hypoglycemic ingredients.

## Figures and Tables

**Figure 1 molecules-27-05360-f001:**
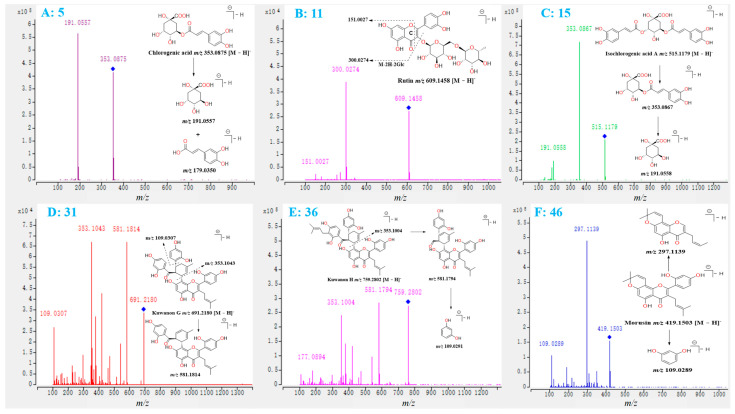
The MS/MS spectra of standards **5** (**A**), **11** (**B**), **15** (**C**), **31** (**D**), **36** (**E**), and **46** (**F**), and corresponding fragmentation pathways.

**Figure 2 molecules-27-05360-f002:**
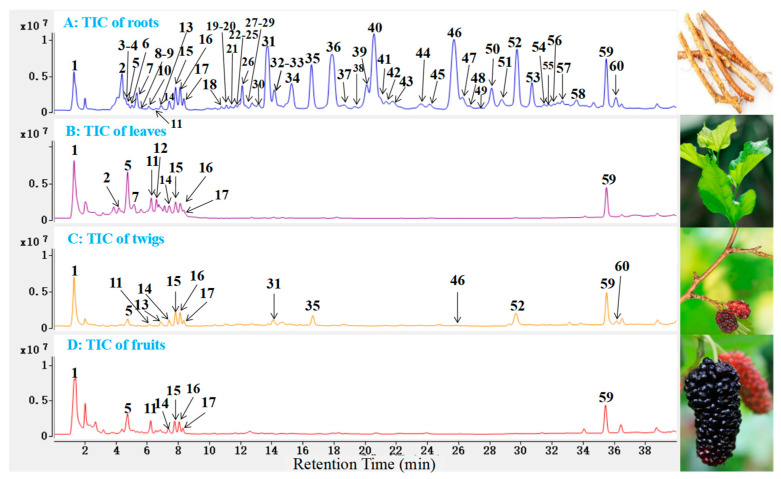
TICs of *M. alba* roots (**A**), leaves (**B**), twigs (**C**), and fruits extracts (**D**), and the peaks of metabolites **1**–**60**. 70% ethanol was used as the extraction solvent.

**Figure 3 molecules-27-05360-f003:**
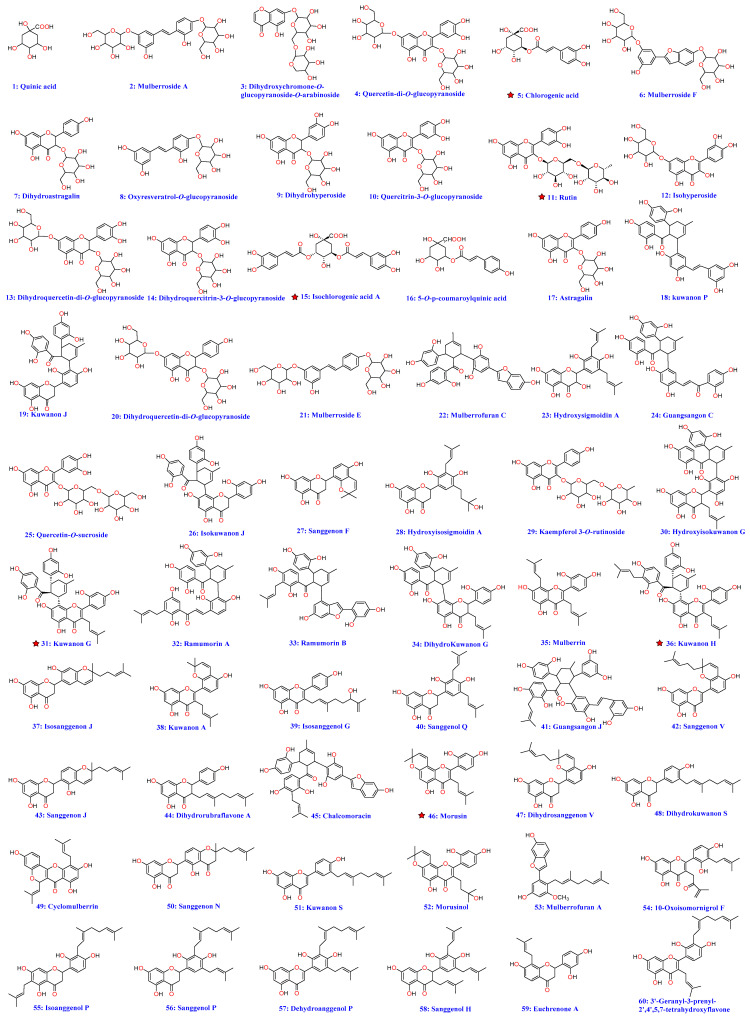
Structures of identified metabolites **1**–**60** (The red star represents the standards).

**Figure 4 molecules-27-05360-f004:**
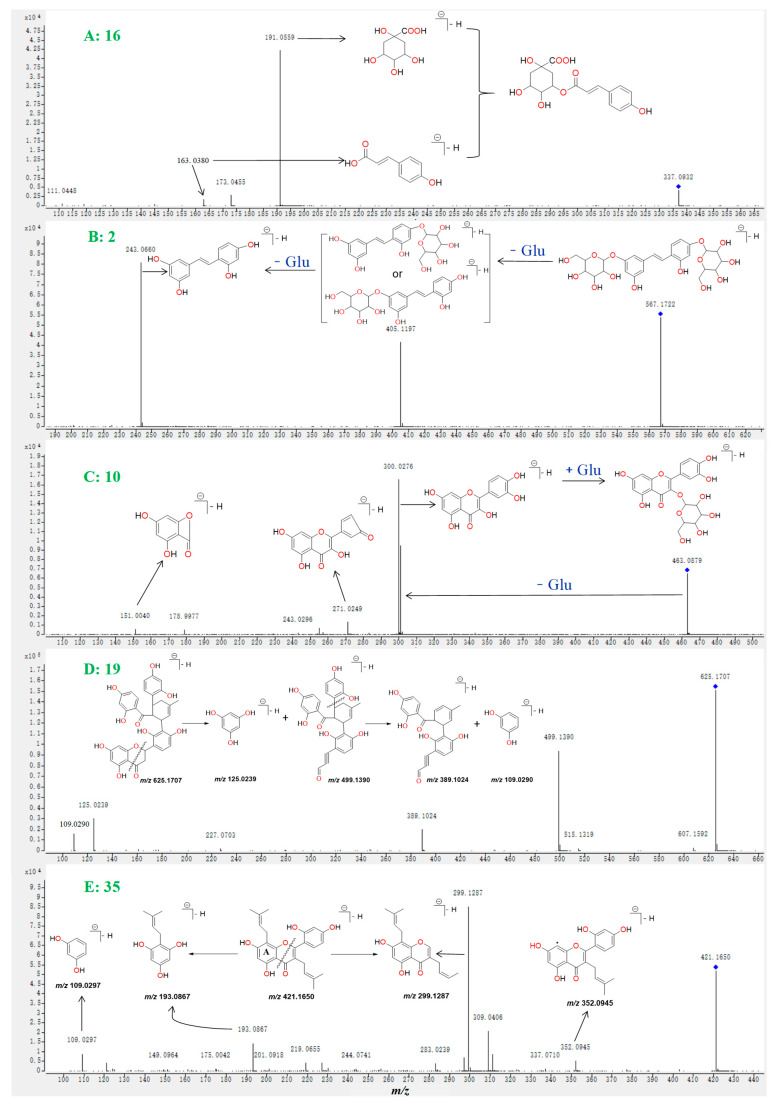
Identification compounds **16** (**A**), **2** (**B**), **10** (**C**), **19** (**D**), and **35** (**E**) based on their MS/MS spectra and the corresponding characteristic fragment ions.

**Figure 5 molecules-27-05360-f005:**
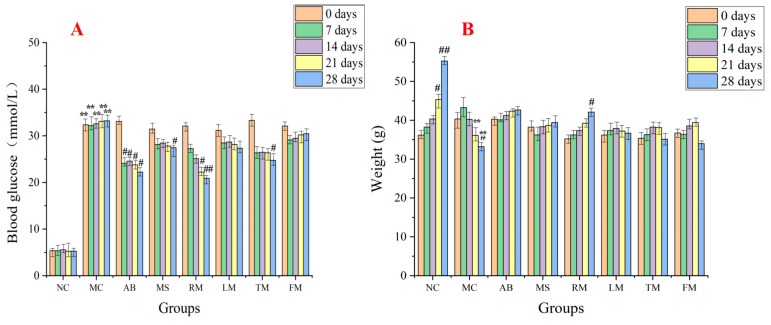
The effect of daily administration of *M. alba* roots, leaves, twigs, and fruits extracts on the blood glucose concentration (**A**) and the body weight variation (**B**) of T2D mice. Data are reported as mean ± SD. For statistical significance, ** *p* < 0.01 compared with the normal control group on the same days. # *p* < 0.05, ## *p* < 0.01 compared with the data at the 0 days. NC: normal group; MC: model group; AB: acarbose group (positive control, 50 mg/kg); MS: morusin group (50 mg/kg); RM: roots of *M. alba* group (200 mg/kg); LM: leaves of *M. alba* group (200 mg/kg); TM: twigs of *M.*
*alba* group (200 mg/kg); FM: fruits of *M. alba* group (200 mg/kg).

**Figure 6 molecules-27-05360-f006:**
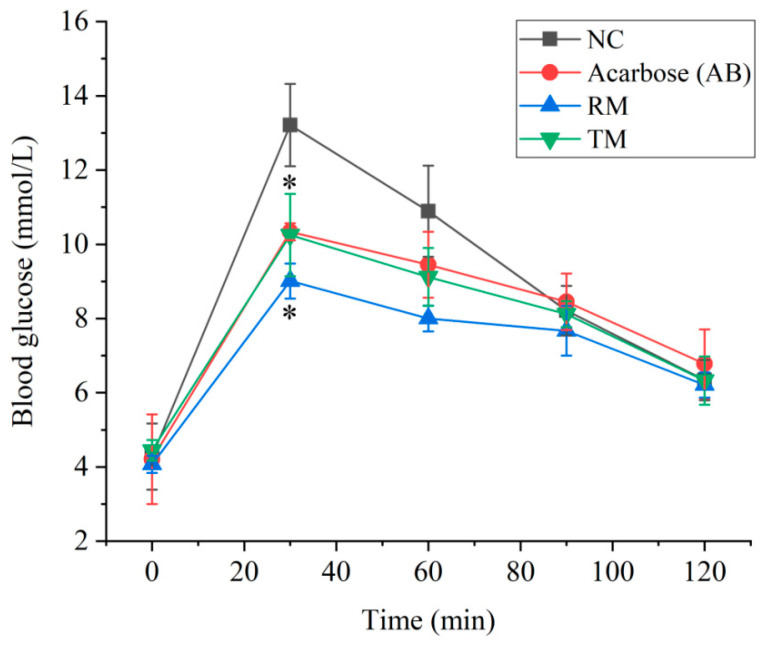
Postprandial glycemia level of normal mice after administration of sucrose, acarbose, roots, and twig extracts. * *p* < 0.05 compared with the normal control group at the 30 min. NC: normal group; AB: acarbose group (positive control, 50 mg/kg); RM: roots of *M. alba* group (200 mg/kg); TM: twigs of *M. alba* group (200 mg/kg).

**Table 1 molecules-27-05360-t001:** Phytochemical compounds identified from different parts of *M. alba* by HPLC-Q-TOF-MS.

No.	*t*_R_ (min.)	[M − H] (*m*/*z*)	Error (ppm)	Molecular Formula	Belongs	MS/MS Fragment Ions (*m*/*z*)	Identification
**1** * ^a^ *	1.29	191.0562	3.4	C_7_H_12_O_6_	All	173.0432, 111.0095	Quinic acid
**2**	4.38	567.1722	−1.4	C_26_H_23_O_14_	All	405.1197, 243.0660	Mulberroside A
**3** * ^a^ *	4.58	471.1170	6.5	C_20_H_24_O_13_	R, T	177.0190	Dihydroxychromone-*O*-glucopyranoside-*O*- arabinoside
**4**	4.66	625.1460	4.8	C_27_H_29_O_17_	R, T	463.0908, 299.0190, 151.0020	Quercetin-di-*O*- glucopyranoside
**5** * ^c^ *	4.73	353.0877	1.1	C_16_H_18_O_9_	All	191.0559, 173.0450, 135.0447	Chlorogenic acid
**6**	5.03	565.1577	3.5	C_26_H_30_O_14_	All	419.2454, 403.1031, 241.0511	Mulberroside F
**7** * ^a^ *	5.32	449.1075	−2.0	C_21_H_22_O_11_	R, L, T	287.0557, 151.0022, 125.0237	Dihydroastragalin
**8** * ^a^ *	5.61	405.1198	2.9	C_20_H_22_O_9_	R	243.0661, 175.0754	Oxyresveratrol-*O*- glucopyranoside
**9** * ^a^ *	5.69	465.1024	−1.9	C_21_H_22_O_12_	R	437.1070, 303.0508, 125.0234	Dihydrohyperoside
**10** * ^a^ *	6.11	463.0879	0.4	C_21_H_20_O_12_	All	350.0276, 271.0249, 151.0040	Quercetin-3-*O*- glucopyranoside
**11** * ^c^ *	6.23	609.1462	0.9	C_27_H_30_O_16_	All	300.0277, 271.0245, 151.0034	Rutin
**12** * ^a^ *	6.58	463.0863	−3.0	C_21_H_20_O_12_	All	301.0347, 151.0030	Isohyperoside
**13**	6.88	627.1557	−0.6	C_27_H_32_O_17_	All	465.1017, 447.0923, 285.0391	Dihydrouercetin-di- O-glucopyranoside
**14** * ^a^ *	7.34	465.1079	3.6	C_21_H_22_O_12_	All	437.1108, 303.0511, 125.0245	Dihydroquercetin- 3-*O*-glucopyranoside
**15***^a^*,^*c*^	7.82	515.1177	−2.5	C_25_H_24_O_12_	All	353.0866, 191.0552, 173.0433	Isochlorogenic acid A
**16** * ^a^ *	8.11	337.0932	3.2	C_16_H_18_O_9_	All	191.0559, 173.0455	5-*O*-p-coumaroylquinic acid
**17**	8.36	447.0901	−6.0	C_21_H_20_O_11_	All	401.1437, 269.1012	Astragalin
**18**	10.77	581.1815	0.5	C_34_H_30_O_9_	R, T	471.1815, 361.1074, 109.0294	kuwanon P
**19** * ^a^ *	11.08	625.1707	−0.4	C_35_H_30_O_11_	R	499.1390, 389.1024, 125.0289	Kuwanon J
**20** * ^b^ *	11.10	647.1407	4.3	C_27_H_32_O_16_	All	611.1599, 449.1100, 125.0248	Dihydroquercetin-di- *O*-glucopyranoside
**21** * ^b^ *	11.35	597.1814	−0.8	C_26_H_32_O_13_	R	551.1761, 389.1240, 227.0711	Mulberroside E
**22**	11.68	579.1656	0.1	C_34_H_28_O_9_	R	469.1274, 359.0919, 109.0290	Mulberrofuran C
**23** * ^a^ *	11.69	439.1758	0.2	C_25_H_28_O_7_	R	421.1653, 313.1435, 125.1236	Hydroxysigmoidin A
**24**	11.70	609.1758	−0.4	C_35_H_30_O_10_	R	499.1414,447.1418, 337.1065, 227.0726, 161.0234	Guangsangon C
**25** * ^a^ *	11.82	625.1429	−3.8	C_27_H_30_O_17_	All	463.0882, 299.0183, 125.0234	Quercetin-*O*-sucroside
**26** * ^a^ *	12.09	625.1705	0.6	C_35_H_30_O_11_	R	499.1393,389.1025, 279.0661, 125.0232, 109.0292	Isokuwanon G
**27**	12.30	353.1036	3.1	C_20_H_18_O_6_	R, T	177.0210, 125.0236	Sanggenon F
**28** * ^a^ *	12.32	439.1764	−2.3	C_26_H_30_O_7_	R, T	287.1600, 151.0014, 125.0243	Hydroxyisosigmoidin A
**29**	12.4	593.1530	3.8	C_27_H_30_O_15_	All	284.0333, 125.0236	Kaempferol 3-*O*-rutinoside
**30** * ^a^ *	13.10	709.2278	−0.9	C_40_H_38_O_12_	R	583.1965,473.1587, 445.1647, 125.0236, 109.0285	Hydroxyisokuwanon G
**31** * ^c^ *	13.71	691.2178	−0.2	C_40_H_36_O_11_	R, T	581.1812, 419.1485, 353.1019	Kuwanon G
**32**	14.16	677.2390	0.4	C_40_H_38_O_10_	R	515.2072, 337.1086, 161.0244	Ramumorin A
**33**	14.20	647.2278	−0.4	C_39_H_36_O_9_	R	537.1910, 427.1505, 109.0290	Ramumorin B
**34** * ^a^ *	15.29	693.2328	−1.1	C_40_H_38_O_11_	R, T	567.2013, 389.1021, 125.0239	Dihydrokuwanon G
**35**	16.62	421.1650	−0.2	C_25_H_26_O_6_	R, T	352.0945, 299.1287, 109.0297	Mulberrin
**36** * ^c^ *	17.89	759.2809	0.3	C_45_H_44_O_11_	R	581.1824, 353.1033, 109.0290	Kuwanon H
**37** * ^a^ *	18.62	421.1643	−1.9	C_25_H_26_O_6_	R, T	269.1535, 151.0028, 125.0237	Isosanggenon J
**38** * ^c^ *	19.36	419.1496	0.2	C_25_H_24_O_6_	R	231.0660, 151.0041, 125.0240	Kuwanon A
**39** * ^a^ *	20.10	421.1648	−0.7	C_25_H_26_O_6_	R, T	309.0400, 231.0659, 125.0240	Isosanggenol G
**40**	20.57	423.1814	1.4	C_25_H_28_O_6_	R, T	395.1864, 379.1919, 125.0244	Sanggenol Q
**41**	20.95	759.2812	0.7	C_45_H_44_O_11_	R	581.1803, 471.1434, 419.1505 379.1141, 353.1007, 109.0290	Guangsangon J
**42**	21.34	419.1495	0.0	C_25_H_24_O_6_	R	231.0659, 151.0027, 125.0233	Sanggenon V
**43**	21.81	421.1643	−1.9	C_25_H_26_O_6_	R	151.0028, 125.0237	Sanggenon J
**44** * ^a^ *	23.62	407.1854	−1.2	C_25_H_28_O_5_	R	389.1764, 297.1487, 109.0289	Dihydrorubraflavone A
**45**	24.15	647.2296	−2.3	C_39_H_36_O_9_	R	469.1299, 359.0928, 109.0291	Chalcomoracin
**46**	25.72	419.1500	1.1	C_25_H_24_O_6_	R, T	297.1134, 191.0709, 109.0291	Morusin
**47** * ^a^ *	26.25	421.1639	−2.8	C_25_H_26_O_6_	R, T	295.1333, 151.0028, 125.0239	Dihydrosanggenon V
**48** * ^a^ *	26.72	407.1883	5.8	C_25_H_28_O_5_	R	407.1883, 283.0635, 137.0265	Dihydrokuwanon S
**49**	27.16	419.1505	1.4	C_25_H_24_O_6_	R, T	263.0893, 203.0744	Cyclomulberrin
**50**	28.11	437.1599	−0.4	C_25_H_26_O_7_	R	368.0899, 151.0037, 125.0245	Sanggenon N
**51**	28.74	405.1708	1.4	C_25_H_26_O_5_	R	335.0920, 321.0749, 282.0525	Kuwanon S
**52**	29.73	437.1641	9.1	C_25_H_26_O_7_	R, T	381.1001, 365.1049, 309.0430	Morusinol
**53**	30.72	391.1943	−0.2	C_25_H_28_O_4_	R	323.1339, 267.0676, 149.0259	Mulberrofuran A
**54**	31.48	435.1449	2.2	C_25_H_24_O_7_	R	366.0742, 151.0032, 125.0242	10-oxoisomornigrol F
**55**	32.09	491.2400	−6.9	C_30_H_36_O_6_	R	297.1490, 193.0865	Isoanggenol P
**56**	32.33	491.2388	1.4	C_30_H_36_O_6_	R	463.2458, 193.0854, 177.0176	Sanggenol P
**57** * ^a^ *	32.66	489.2290	−2.6	C_33_H_40_O_19_	R	433.1634, 377.1048	Dehydroanggenol P
**58**	33.58	491.2422	−2.4	C_30_H_36_O_6_	R	473.2317, 365.2116, 125.0236	Sanggenol H
**59**	35.48	339.1235	1.5	C_20_H_20_O_5_	All	177.0919, 161.0243, 135.0450	Euchrenone A
**60**	36.11	489.2288	2.2	C_30_H_34_O_6_	R, T	231.0656, 216.0383, 179.0587	3′-Geranyl-3-prenyl- 2′,4′,5,7-tetrahydro- xyflavone

*^a^* The compound was reported for the first time from *M. alba*. *^b^* The [M – H + HCOO]**^−^**. *^c^* The compound was unambiguously determined by comparing the *t*_R_, MS and MS/MS with the references.

## Data Availability

Not applicable.

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
