# Peer review of "LC-MS Metabolite Profiling and the Hypoglycemic Activity of Morus alba L. Extracts"

_molecules, 2022, doi:10.3390/molecules27175360_

Round 1
Reviewer 1 Report
Materials and methods- Line 266- Abbreviation should precede with full word- ICR
Line 274- Please include citation/ref. for the blood glucose level to determine diabetic mice.
Line 284- Please state the software used and country
Conclusion- Line 289- Numbers less than 10 should be written in words.
Line 296-mouse model
Author Response
Reviewer 1:
- Materials and methods- Line 266- Abbreviation should precede with full word- ICR
Response:
We thank your professional suggestion. We have made the corresponding correction in the revised manuscript (The full words of “Institute of Cancer Research” was added to ICR).
- Line 274- Please include citation/ref. for the blood glucose level to determine diabetic mice.
Response:
The corresponding reference has been added to determine the diabetic mice. The specific detail is as follow:
- Line 284- Please state the software used and country
Response:
The software used and the country has been added to the revised manuscript (SPSS 16.0 statistics software (Inc., Chicago, USA) was used for experimental data statistical analysis).
- Conclusion- Line 289- Numbers less than 10 should be written in words.
Response:
We thank your valuable suggestions. We have made corresponding corrections in the entire revised manuscript
- Line 296-mouse model
Response:
All authors thank your careful review, the corresponding mistake has been corrected.

Reviewer 2 Report
Review Zhou et al
Zhou and colleagues identify the phytocomponents present in extracts from roots, leaves, twigs and fruit from Morus alba L and show hypoglycemic effects of root and twig extracts in hyperglycemic ICR mice.
The findings are interesting. Unfortunately, the authors have only investigated blood glucose levels (after at least 7 days of administration), so there is no indication whatsoever about a possible target of the hypoglycemic extracts, nor any speculation on that issue. In contrast to the tremendous efforts of the authors on the identification of the phytocompounds, the biological evaluation of the extracts containing those compounds is quite limited. Several of my concerns concern this issue.
MAJOR ISSUES
1. Data and speculation on the hypoglycemic effects are scarce and lacking, respectively. The authors should at least mention the possible mechanisms by which the extracts may exert their hypoglycemic effect in the Discussion section.
2. Related to point 1, it would have been interesting to inject the extracts and measure their possible acute (0 to 120 minutes) effect on blood glucose in hyperglycemic mice.
Alternatively, performing a GTT in normal mice with the extracts injected 30 minutes before the glucose may be interesting.
Any of these two experiments could provide a clue to the mechanism by which the extracts reduce glycemia.
3. This point is related to point 1, where the authors do not speculate on the possible mechanism(s) by which their extracts exert their hypoglycemic effect: the authors describe that an anti-diabetic activity of their plant has been described before in previous studies and mention (only) 3 articles, references 22-24. The first one (ref 22), concerns another plant, not the Morus alba. While the plant in ref 22 is somewhat related to the plant studied in the current study, this is a very serious shameful mistake. The other two references are reviews, so do not contain original findings. Anyway, mentioning only 3 references (without giving any insight into whatever is mentioned in these papers) gives the idea that not so much is known about the effect of Morus alba extracts on hyperglycemia and other events in diabetes and that the contents of the current manuscript is therefore very new. However, PubMed alone reveals already 150 papers on Morus alba/mulberry & diabetes. This makes it impossible for any reader or a reviewer to acknowledge the novelty of the findings of the current manuscript (unless the reader/reviewer is already an expert in [mulberry vs diabetes].
This reviewer wishes that the authors cite several (10-20) important original articles (no reviews) of the 150 papers mentioned above (thus preventing that it will look like a review on the subject), in order to:
1. put the current manuscript in (a better) perspective of what is already known on the subject and to clearly DEMONSTRATE the novelty of this study (instead of just telling that it is new and better)
2. use the incredible amount of knowledge already present on the subject (in literature) to try to explain the hypoglycemic effects of their root and twig extracts (see major point 1).
Just ignoring of what others have already demonstrated is just not very scientific and should be avoided.
4. This issue may not concern the authors but especially the production managers of the journal. Several figures are too small to be read by the naked eye. It's of no interest to publish manuscripts with figures that have drawings or text that cannot be read, interpreted and appreciated by the reader. Moreover, these figures are even too small to be reviewed by this reviewer. This concerns figures 1 and 3. Please increase the size of these 2 figures.
MINOR ISSUES
1. Throughout the manuscript, including the title, the authors mention 'anti-diabetic activities'. However, the authors have only investigated one single diabetic activity, which is potential hypoglycemic effect, after mid-long term administration. Throughout the manuscript, 'activities' should be changed into 'activity', including title.
2. Manuscript title is a bit awkward. I suggest 'LC-MS metabolite profiling and anti-diabetic activity of Morus alba L extracts'
3. Line 13 (abstract): please replace 'anti-diabetic activities' for 'hypoglycemic activity' or 'blood glucose-lowering activity'. 'Anti-diabetic activities'' is too vague, not precise enough, and multiple (see minor point 1)
4. Throughout the manuscript (about 10 times in total), the authors mention 'THE Type 2 diabetes mellitus mouse model'. First of all, 'their' model cannot be considered as 'the' model, since there are multiple T2D mouse models. The authors should avoid to mention 'THE T2D model', since it implies that there exists only one single model. Second, their model is only explained in the Materials & Methods section. Their model should be clear from the start. I suggest to mention in line 13 (abstract) '..... evaluated in the hyperglycemic high fat diet/streptozotocin T2D mouse model'.
5. Line 19. In order to improve readability, finish the phrase with '... in contrast to the extracts of leaves and fruits.'
6. One has to analyze the blood glucose graph in order to appreciate the quantitative effects of the extracts (and of morusin). In the text (both abstract and results) it is only indicated that glycemia was lower. Please state important findings quantitatively in both Results and Abstract sections. For example: 'After 4 weeks of treatment with root and twig extracts, glycemia was significantly reduced from ... +/- ... to ... +/- ... mM and from .... +/- ... to ... +/- ... mM, respectively.' Then mention the same for acarbose and morusin to put the extract data in perspective. This will not only supply the reader with important information, it will also increase the significance of the data and the interest of the manuscript (especially when people are screening potentially interesting manuscripts on PubMed).
7. Final phrase of Abstract : authors claim that their work is a 'COMPREHENSIVE investigation of anti-diabetic activities', which is not the case. Please rephrase this sentence.
8. Final phrase of Abstract: authors claim that their study 'provides a template for other medicinal plants'. This reviewer does not have the same feeling. Authors should not overestimate their work. Please remove this part of the final phrase.
9. In the Discussion section, the authors concentrate on compounds 31, 36, 40, 46 and mention that these compounds were 'regarded as the potential anti-diabetic metabolites'. This reviewer does not share this assumption and finds that this phrase is not justified. This phrase could be replaced by '..., and we consider the high-level prenylated flavonoids, such as compounds ..... as potential anti-diabetic metabolites' (in this way it is clear that it is their personal (un-objective) opinion).
10. In their discussion, the authors are exclusively concentrating on the root extract. Unfortunately, this extract is extremely rich in metabolites which makes is virtually impossible to state which metabolites are responsible for its hypoglycemic effect (see also minor point 9). In sharp contrast, the twig extract is lowering glycemia as well, but only has few phytometabolites that are absent in leave and fruit extracts (which do not lower glycemia), in particular compounds 36 and 52. These two compounds have a higher probability to lower blood glucose than the compounds mentioned by the authors (see minor point 9). These two compounds should be mentioned at the end of the Results/Discussion section and again in the Conclusion section. Of interest, compound 52 is morusinol, largely related to morusin (46) which is already somewhat active in reducing glycemia.
11. Isn't metabolite number 16 also quite visible in leaves, twigs and fruits? The same for number 14 in fruit extract? If this is the case, please add these numbers in the respective TIC graphs.
12. In line 190 it is stated that the plant parts were extracted by ethanol while in line 247 it is stated that extracts were prepared using methanol. This is worrying. Ethanol or methanol?
13. It should be more clear from the beginning what the extracts truly are. Please mention in the Abstract and in the legend of Fig 2 that the extracts are methanolic or 70% methanol extracts (or ethanol see point 12).
14. Figure 5 also shows glycemia on day 21. In the text, this day (and only this day) is left out. Why? Reintegrate this day in lines 193, 195, and 198.
15. Line 207: data are presented as mean +/- SD. Line 285: data are presented as mean +/- SEM. These conflicting statements are extremely worrying. SD or SEM?
16. This reviewer would also like to see ** significance in Fig 5B (MC vs NC)
17. Line 218: authors claim that the increase in body weight indicated a reversal of diabetes. This conclusion cannot be drawn. This should be tuned down into: '.... displayed significant activity to reverse the T2D-associated weight loss in diabetic mice'.
18. Line 225: ' which indicated that morusin was one of the anti-diabetic ingredients of...'. is an overstatement. It should be changed into 'which indicated the presence of other compounds with hypoglycemic activity in root extract'. Lines 301-303 display the same problem. Please also change here.
19. Line 227. Again an overstatement. Please change 'demonstrated' into 'could suggest'.
20. Paragraph 3.4. line 266. How old were the mice? (at the beginning of treatment).
21. Same paragraph: mice were males or females?
22. Paragraph 3.5 line 273. It is not clear if the NC mice were also fed a high fat diet. If these mice were fed a normal diet (not high fat), then 'In addition' should be replaced by 'In contrast'. Looking at graph 5B is seems clear to me that those NC mice were on the same high fat diet (mice of 55 grams!), but is should be clearly stated in the text of paragraph 3.5.
23. Same paragraph: HFD was during 30 days. Mice were injected STZ for 3 days. But when? Before the high fat diet or in the middle? At which day?
24. The experiment was initiated with 90 mice, but only 80 mice were used in the groups. Does this mean that glycemia was tested before extract treatment to discard mice that were outside of a certain glycemic range?
25. Line 274: replace 'glucose level' by 'blood glucose level'
26. Same paragraph: in what vehicle the extracts were dissolved in order to administrate the extracts? Please describe.
27. Same paragraph: how were the extracts administrated? Intragastrically, as for the 2 control groups? Please describe.
28. Line 280: 'for successive 28 days'. I assume that this means that mice were treated daily during 28 days? If this is the case, please change into '....of 20 mL/kg. Mice were treated daily during 28 days'. Also, please mention in the Results section, legend of Fig 5: 'The effect of daily administration of M. alba root, leaf, twig, and fruit extracts on ......'
Author Response
Reviewer 2:
Review Zhou et al
Zhou and colleagues identify the phytocomponents present in extracts from roots, leaves, twigs and fruit from Morus alba L and show hypoglycemic effects of root and twig extracts in hyperglycemic ICR mice.
The findings are interesting. Unfortunately, the authors have only investigated blood glucose levels (after at least 7 days of administration), so there is no indication whatsoever about a possible target of the hypoglycemic extracts, nor any speculation on that issue. In contrast to the tremendous efforts of the authors on the identification of the phytocompounds, the biological evaluation of the extracts containing those compounds is quite limited. Several of my concerns concern this issue.
MAJOR ISSUES
- Data and speculation on the hypoglycemic effects are scarce and lacking, respectively. The authors should at least mention the possible mechanisms by which the extracts may exert their hypoglycemic effect in the Discussion section.
Response:
The possible mechanisms by which the extracts may exert their hypoglycemic effect were added in the Discussion section as follow:
Four main mechanisms of hypoglycemic activity for the M. alba roots, twigs, or leaves extracts were investigated in previous studies. The first one was inhibiting the intestinal α-glucosidase activity, thereby blocking the supply of glucos. The second was improving glucose uptake and promoting glycogen synthesis by activating the corresponding signaling pathway or protein. The third was inhibiting oxidative stress, apoptosis, and inflammation and protecting the pancreatic islet cells. The last one was decreasing the level of total cholesterol, triglyceride, and low-density lipoprotein cholesterol and increasing the high-density lipoprotein cholesterol further to reduce the blood glucose content. In this study, the extracts of M. alba roots and twigs display significant hypoglycemic activity and reduced postprandial glycemia function. The postprandial glycemia experiment results (Supplementary experiment for this studies) indicated that inhibiting the intestinal α-glucosidase activity to block the supply of glucose was the potential mechanism of hypoglycemic activity. In addition, the metabolites 11 (Rutin), 31 (Kuwanon G) and 46 (Morusin) identified from the extracts of M. alba roots and twigs were α-glucosidase inhibitors according to previous reports. Therefore, inhibiting the intestinal α-glucosidase activity and thereby reducing the postprandial glycemia was the primary mechanism of hypoglycemic activity for the M. alba roots and twigs.
- Related to point 1, it would have been interesting to inject the extracts and measure their possible acute (0 to 120 minutes) effect on blood glucose in hyperglycemic mice.
Alternatively, performing a GTT in normal mice with the extracts injected 30 minutes before the glucose may be interesting.
Any of these two experiments could provide a clue to the mechanism by which the extracts reduce glycemia.
Response:
It is difficult to inject the extracts and measure their effect on blood glucose in hyperglycemic mice because this experiment is more than 40 days. Therefore, the experiment on postprandial glycemia of normal mice has been added to this study to investigate the possible mechanism.
Experimental procedure:
Twenty-four normal mice (4 weeks old, weight range of 18.0-22.0 g) were randomly divided into four groups based on their weight, each containing six mice. All mice were given sucrose solution with 2.0 g/kg. In addition to the normal control group (NC), the other three groups were given acarbose (50 mg/kg), roots and twigs extracts solution (200 mg/kg for RM and TM, respectively). The blood glucose level was determined by the glucometer at 0, 30, 60, 90, and 120 min.
Experimental Results:
The result shows that the acarbose (10.34 ± 0.23 mmol/L), roots (9.01 ± 0.47 mmol/L) and twigs extracts (10.25 ± 1.11 mmol/L) groups display significant activity in reducing postprandial glycemia (P < 0.05) comparing to the normal control group (13.21 ± 1.11 mmol/L) after administration of corresponding solution for 30 min (Table R1). The extracts of M. alba roots display stronger activity than the positive control (AB) and the twigs extracts (TM) for reducing postprandial glycemia. The blood glucose level of all groups returns to baseline after 120 min.
Table R1 The postprandial glycemia (mmol/L) of the normal mice.
|
Time (min) |
blood glucose (mmol/L) |
|||
|
NC |
Acarbose (AB) |
RM |
TM |
|
|
0 |
4.28 ± 0.89 |
4.21 ± 1.21 |
4.07 ± 0.23 |
4.45 ± 0.28 |
|
30 |
13.21 ± 1.11 |
10.34 ± 0.23* |
9.01 ± 0.47* |
10.25 ± 1.11* |
|
60 |
10.89 ± 1.23 |
9.45 ± 0.89 |
8.00 ± 0.35 |
9.12 ± 0.78 |
|
90 |
8.21 ± 0.67 |
8.45 ± 0.76 |
7.67 ± 0.67 |
8.12 ± 0.35 |
|
120 |
6.34 ± 0.54 |
6.78 ± 0.93 |
6.21 ± 0.34 |
6.33 ± 0.65 |
* p < 0.05 compared with the normal control group (NC). NC: normal group; AB: acarbose group (positive control, 50 mg/kg); RM: roots of M. alba group (200 mg/kg); TM: twigs of M. alba group (200 mg/kg).
Figure R1. The postprandial glycemia level of normal mice after administration of acarbose, roots, and twigs extracts. * p < 0.05 compared with the normal control group at the 30 min. NC: normal group; AB: acarbose group (positive control, 50 mg/kg); RM: roots of M. alba group (200 mg/kg); TM: twigs of M. alba group (200 mg/kg).
Potential mechanism:
The postprandial glycemia experiment results indicated that inhibiting the intestinal α-glucosidase activity to block the supply of glucose was the potential mechanism of hypoglycemic activity. In addition, the metabolites 11 (Rutin), 31 (Kuwanon G) and 46 (Morusin) identified from the extracts of M. alba roots and twigs were α-glucosidase inhibitors according the previous reports. Therefore, inhibiting the intestinal α-glucosidase activity, consequently blocking the supply of glucose and reducing the postprandial glycemia, was the primary mechanism of hypoglycemic activity for the M. alba roots and twigs.
The above content have been added in the revised manuscript.
- This point is related to point 1, where the authors do not speculate on the possible mechanism(s) by which their extracts exert their hypoglycemic effect: the authors describe that an anti-diabetic activity of their plant has been described before in previous studies and mention (only) 3 articles, references 22-24. The first one (ref 22), concerns another plant, not the Morus alba. While the plant in ref 22 is somewhat related to the plant studied in the current study, this is a very serious shameful mistake. The other two references are reviews, so do not contain original findings. Anyway, mentioning only 3 references (without giving any insight into whatever is mentioned in these papers) gives the idea that not so much is known about the effect of Morus alba extracts on hyperglycemia and other events in diabetes and that the contents of the current manuscript is therefore very new. However, PubMed alone reveals already 150 papers on Morus alba/mulberry & diabetes. This makes it impossible for any reader or a reviewer to acknowledge the novelty of the findings of the current manuscript (unless the reader/reviewer is already an expert in [mulberry vs diabetes].
This reviewer wishes that the authors cite several (10-20) important original articles (no reviews) of the 150 papers mentioned above (thus preventing that it will look like a review on the subject), in order to:
- put the current manuscript in (a better) perspective of what is already known on the subject and to clearly DEMONSTRATE the novelty of this study (instead of just telling that it is new and better)
- use the incredible amount of knowledge already present on the subject (in literature) to try to explain the hypoglycemic effects of their root and twig extracts (see major point 1).
Just ignoring of what others have already demonstrated is just not very scientific and should be avoided.
Response:
We thank your valuable and professional suggestions. The authors carefully studied the 142 pieces of literature, which were screened by the keywords Morus alba/mulberry & diabetes in PubMed recommended by the reviewer. The chemical components and hypoglycemic activity of Morus alba L single part extract (Roots, Leaves, Twigs, or Fruits), and the corresponding anti-diabetic mechanisms have been reported in previous studies. However, the comparative evaluation of the hypoglycemic activity and analysis of the primary active metabolites of four different parts of Morus alba L simultaneously were rarely reported.
The references 22-24 were cited irrelevantly, which was replaced by other three closely literature in the revised manuscript. In addition, in order to make readers acknowledge this article extensively and in-depth, eight (26-33) other pieces of literature were cited in the revised manuscript.
The potential possible mechanisms of the extracts that may exert their hypoglycemic effect were added in the Discussion section.
- This issue may not concern the authors but especially the production managers of the journal. Several figures are too small to be read by the naked eye. It's of no interest to publish manuscripts with figures that have drawings or text that cannot be read, interpreted and appreciated by the reader. Moreover, these figures are even too small to be reviewed by this reviewer. This concerns figures 1 and 3. Please increase the size of these 2 figures.
Response:
The author apologizes for the small figures. The size of all figures in the revised manuscript has been increased enough to be read by the naked eye.
MINOR ISSUES
- Throughout the manuscript, including the title, the authors mention 'anti-diabetic activities'. However, the authors have only investigated one single diabetic activity, which is potential hypoglycemic effect, after mid-long term administration. Throughout the manuscript, 'activities' should be changed into 'activity', including title.
Response:
We have made corresponding corrections in the revised manuscript (Each 'activities' has been checked in the whole manuscript and some of them have been revised to 'activity' based on your suggestion).
- Manuscript title is a bit awkward. I suggest 'LC-MS metabolite profiling and anti-diabetic activity of Morus alba L extracts'
Response:
We thank your professional suggestions. We have made corresponding corrections in the revised manuscript (The title has been changed to “LC-MS metabolite profiling and the hypoglycemic activity of Morus alba L. extracts”).
- Line 13 (abstract): please replace 'anti-diabetic activities' for 'hypoglycemic activity' or 'blood glucose-lowering activity'. 'Anti-diabetic activities'' is too vague, not precise enough, and multiple (see minor point 1)
Response:
We thank your professional suggestions. We have made corresponding corrections in the revised manuscript (The 'anti-diabetic activities' was replaced by 'hypoglycemic activity' in the Abstract part and other parts).
- Throughout the manuscript (about 10 times in total), the authors mention 'THE Type 2 diabetes mellitus mouse model'. First of all, 'their' model cannot be considered as 'the' model, since there are multiple T2D mouse models. The authors should avoid to mention 'THE T2D model', since it implies that there exists only one single model. Second, their model is only explained in the Materials & Methods section. Their model should be clear from the start. I suggest to mention in line 13 (abstract) '..... evaluated in the hyperglycemic high fat diet/streptozotocin T2D mouse model'.
Response:
We thank your valuable and professional suggestions. The T2D model was the high-fat diet/streptozotocin-induced T2D mouse model. We have made the corresponding correction in the revised manuscript (‘the high-fat diet/streptozotocin-induced T2D mouse model’ was used in the Abstract part).
- Line 19. In order to improve readability, finish the phrase with '... in contrast to the extracts of leaves and fruits.'
Response:
We have made corresponding correction in the revised manuscript. This sentence were revised as 'In contrast to the extracts of M. alba leaves and fruits, the extracts of roots and twigs displayed significant hypoglycemic activity'.
- One has to analyze the blood glucose graph in order to appreciate the quantitative effects of the extracts (and of morusin). In the text (both abstract and results) it is only indicated that glycemia was lower. Please state important findings quantitatively in both Results and Abstract sections. For example: 'After 4 weeks of treatment with root and twig extracts, glycemia was significantly reduced from ... +/- ... to ... +/- ... mM and from .... +/- ... to ... +/- ... mM, respectively.' Then mention the same for acarbose and morusin to put the extract data in perspective. This will not only supply the reader with important information, it will also increase the significance of the data and the interest of the manuscript (especially when people are screening potentially interesting manuscripts on PubMed).
Response:
The quantitative data on blood glucose has been added in the Abstract part and the Result section. The original data have been provided as Table S2 and S3 in the Supporting Materials.
- Final phrase of Abstract: authors claim that their work is a 'COMPREHENSIVE investigation of anti-diabetic activities', which is not the case. Please rephrase this sentence.
Response:
This sentence has been rewritten in the revised manuscript.
- Final phrase of Abstract: authors claim that their study 'provides a template for other medicinal plants'. This reviewer does not have the same feeling. Authors should not overestimate their work. Please remove this part of the final phrase.
Response:
This sentence has been rewritten as “This work marks the first comparison of the metabolites and hypoglycemic activity of M. alba roots, leaves, twigs, and fruits extracts, which provide a foundation for further development of M. alba extracts as anti-diabetic drugs”.
- In the Discussion section, the authors concentrate on compounds 31, 36, 40, 46 and mention that these compounds were 'regarded as the potential anti-diabetic metabolites'. This reviewer does not share this assumption and finds that this phrase is not justified. This phrase could be replaced by '..., and we consider the high-level prenylated flavonoids, such as compounds ..... as potential anti-diabetic metabolites' (in this way it is clear that it is their personal (un-objective) opinion).
Response:
We have made corresponding corrections in the revised manuscript.
- In their discussion, the authors are exclusively concentrating on the root extract. Unfortunately, this extract is extremely rich in metabolites which makes is virtually impossible to state which metabolites are responsible for its hypoglycemic effect (see also minor point 9). In sharp contrast, the twig extract is lowering glycemia as well, but only has few phytometabolites that are absent in leave and fruit extracts (which do not lower glycemia), in particular compounds 36 and 52. These two compounds have a higher probability to lower blood glucose than the compounds mentioned by the authors (see minor point 9). These two compounds should be mentioned at the end of the Results/Discussion section and again in the Conclusion section. Of interest, compound 52 is morusinol, largely related to morusin (46) which is already somewhat active in reducing glycemia.
Response:
We thank your valuable and professional suggestions. Compounds 36 and 52 were emphasized and mentioned at the end of the Results/Discussion section and again in the Conclusion section based on your suggestions. The specific contents are as follows: Metabolites 36 and 52, which were rich in the extracts of roots and twigs and absent in that of leaves and fruits, were further regarded as hypoglycemic ingredients by comprehensive analysis of the level of both compounds in the four extracts and their hypoglycemic activities. In addition, the structure of compound 52 (Morusinol) was highly similar to that of metabolite 46, which indicated that the high-level compound 52 has potential hypoglycemic activity and needs further studies.
- Isn't metabolite number 16 also quite visible in leaves, twigs and fruits? The same for number 14 in fruit extract? If this is the case, please add these numbers in the respective TIC graphs.
Response:
Metabolite 16 was also present in the TICs of leaves, twigs and fruits, and compound 14 existed in the TIC of fruit extract. The number of compounds, which has an obvious peak in the TICs, has been added to the TICs of roots, leaves, twigs and fruits in Figure 2.
- In line 190 it is stated that the plant parts were extracted by ethanol while in line 247 it is stated that extracts were prepared using methanol. This is worrying. Ethanol or methanol?
Response:
The authors are sorry for that obvious mistake and thank the reviewer’s careful and professional review. 70% ethanol was used as the extraction solvent. We have made the corresponding correction in the revised manuscript.
- It should be more clear from the beginning what the extracts truly are. Please mention in the Abstract and in the legend of Fig 2 that the extracts are methanolic or 70% methanol extracts (or ethanol see point 12).
Response:
70% ethanol was used as the extraction solvent. We have made corresponding corrections in the Abstract part and the legend of Figure 2.
- Figure 5 also shows glycemia on day 21. In the text, this day (and only this day) is left out. Why? Reintegrate this day in lines 193, 195, and 198.
Response:
The authors are sorry for the obvious mistakes and thank the reviewer’s careful reminder. The forgotten day 21 has been added to the revised manuscript.
- Line 207: data are presented as mean +/- SD. Line 285: data are presented as mean +/- SEM. These conflicting statements are extremely worrying. SD or SEM?
Response:
SD (Standard deviation) was used in this study, which has been revised in the revised manuscript.
- This reviewer would also like to see ** significance in Fig 5B (MC vs NC)
Response:
The statistical significance between MC and NC has been added to Figure. 5B. The relative content was also added to the Result and Discussion section.
- Line 218: authors claim that the increase in body weight indicated a reversal of diabetes. This conclusion cannot be drawn. This should be tuned down into: '.... displayed significant activity to reverse the T2D-associated weight loss in diabetic mice'.
Response:
We have made the corresponding correction in the revised manuscript.
- Line 225: ' which indicated that morusin was one of the anti-diabetic ingredients of...'. is an overstatement. It should be changed into 'which indicated the presence of other compounds with hypoglycemic activity in root extract'. Lines 301-303 display the same problem. Please also change here.
Response:
We have made the corresponding corrections in the revised manuscript.
- Line 227. Again an overstatement. Please change 'demonstrated' into 'could suggest'.
Response:
We have made the corresponding correction in the revised manuscript.
- Paragraph 3.4. line 266. How old were the mice? (at the beginning of treatment).
Response:
Four weeks old mice (28 days) were purchased from the animal company and fed for a week (7 days) to adapt to the environment. A high-fat diet was provided to those mice for a month (30 days) and an injection of the STZ for three days. After the T2D mouse model was established, the treatment began. Therefore, at the beginning of treatment, the age of the mice was 68 days, which was added in the revised manuscript.
- Same paragraph: mice were males or females?
Response:
The male mice were purchased in this study, which have been added in the revised manuscript.
- Paragraph 3.5 line 273. It is not clear if the NC mice were also fed a high fat diet. If these mice were fed a normal diet (not high fat), then 'In addition' should be replaced by 'In contrast'. Looking at graph 5B is seems clear to me that those NC mice were on the same high fat diet (mice of 55 grams!), but is should be clearly stated in the text of paragraph 3.5.
Response:
The NC mice were not provided with a high-fat diet in this study. The normal diet was given to the NC group mice. The average weight reached approximately 55 grams after feeding for 96 days.
- Same paragraph: HFD was during 30 days. Mice were injected STZ for 3 days. But when? Before the high fat diet or in the middle? At which day?
Response:
The high-fat diet was provided to mice for a month (30 days). After the high-fat diet, the injection of the STZ for 3 days. Namely, the STZ was begun to inject for the 65 days old mice. The corresponding data have been added to the revised manuscript.
- The experiment was initiated with 90 mice, but only 80 mice were used in the groups. Does this mean that glycemia was tested before extract treatment to discard mice that were outside of a certain glycemic range?
Response:
Yes. The blood glucose level of most mice was more than 11.6 mmol/L after intraperitoneal injection of STZ for three days, however, the blood glucose level of several mice was less than 11.6 mmol/L after intraperitoneal injection of STZ for three days. Those mice (Blood glucose level < 11.6 mmol/L) were not given extract treatment. Therefore, only 80 mice were used in the groups.
- Line 274: replace 'glucose level' by 'blood glucose level'
Response:
We have made the corresponding corrections in the revised manuscript.
- Same paragraph: in what vehicle the extracts were dissolved in order to administrate the extracts? Please describe.
Response:
Distilled water was used as a solvent for the extracts, which has been added in the revised manuscript.
- Same paragraph: how were the extracts administrated? Intragastrically, as for the 2 control groups? Please describe.
Response:
All groups were given the corresponding solution by intragastric administration, which has been added to in the revised manuscript.
- Line 280: 'for successive 28 days'. I assume that this means that mice were treated daily during 28 days? If this is the case, please change into '....of 20 mL/kg. Mice were treated daily during 28 days'. Also, please mention in the Results section, legend of Fig 5: 'The effect of daily administration of M. alba root, leaf, twig, and fruit extracts on ......'
Response:
We have made the corresponding corrections in the revised manuscript.

Round 2
Reviewer 2 Report
The manuscript has significantly improved.
Several minor issues (remaining or new ones):
1. Next time, when addressing the points raised by a reviewer, please indicate in the Reply To Reviewer document the new line numbers where the text has been changed, in addition to just highlighting the modified text. This facilitates the work of the reviewer and of the editor and prevents UPSETTING THE REVIEWER. In the current second version of your manuscript, the reviewer needs to go back and look in the first version of the manuscript to see in which part of the manuscript there had been a problem (since (s)he correctly and respectfully indicated the line numbers where there were issues) and then try to refind that text in the second version of the manuscript. This is an incredible loss of time for the reviewer (and editor) and very frustrating.
2. This point is a minor issue which this reviewer had overlooked previously. I apologize. Please state in the Materials and Methods section at which time glycemia was measured concerning figure 5A. Are these fed animals? Measured at the very beginning of the light cycle? End of the light cycle? Had these animals been starved before measuring glycemia?
3. Same for the new figure 6. This reviewer assumes that these animals were probably starved before treatment, but this is stated nowhere.
4. Lines 284-285. It is stated that "compounds 11, 31, and 46 were present in roots and twigs". Then the authors conclude that it is mostly the alpha-glucosidase activity that is the primary mechanism explaining the hypoglycemic effect of these two extracts. This reviewer does not understand this reasoning. First of all, in contrast to what the authors state, the 3 compounds mentioned above are NOT PRESENT in twigs. This is a terrible mistake (and I don't think it is the job of a reviewer to correctly interpret the results of a manuscript). Please correct.
5. Related to issue 4. The resulting conclusion of this error (under 4) is therefore incorrect. Conclusion should be tuned down and be more objective.
I suggest "...inhibiting the intestinal alpha-glucosidase activity and thereby .... could POSSIBLY be a major mechanism responsible for the hypoglycemic activity of M. alba, especially in root extracts. Nevertheless, other mechanisms, for instance those involving an insulin sensitising, insulin mimicking, or insulinotropic action cannot be excluded."
If (also) a GTT had been performed as suggested by this reviewer (previous minor point #2), instead of the sucrose administration, these 3 latter possibilities could possibly have been discarded. Now, these 3 possibilities cannot be ignored.
6. The legend of Fig 6 lacks information. Was sucrose co-administrated with the extracts? (sucrose is not even mentioned in the legend). Orally? Both at time zero?
7. This concerns previous minor issue #24. Apparently, as confirmed by the authors, ten mice with blood glucose levels below 11.6 mM were discarded. This should be stated in line 345, after ".....after the high-fat diet feeding."'
8. This concerns previous minor point #22. As the authors now confirm, the NC mice received normal diet (chow). This should be stated in the text (line 342), because possibly, a control group could have been given the high-fat diet without the STZ, because this would also not lead to hyperglycemia. It's just another type of control.
9. Lines 350-351. During 28 days, mice were treated daily with extracts or controls. During these 28 days, high-fat diet was maintained? If this was the case, please state so.
10. Paragraph 3.6. NC mice received water? If this was the case, please state accordingly. Were extracts/controls co-administrated with sucrose? Intragastrically? Please note these details in the text of this paragraph.
Author Response
#Reviewer 2:
- Next time, when addressing the points raised by a reviewer, please indicate in the Reply To Reviewer document the new line numbers where the text has been changed, in addition to just highlighting the modified text. This facilitates the work of the reviewer and of the editor and prevents UPSETTING THE REVIEWER. In the current second version of your manuscript, the reviewer needs to go back and look in the first version of the manuscript to see in which part of the manuscript there had been a problem (since (s)he correctly and respectfully indicated the line numbers where there were issues) and then try to refind that text in the second version of the manuscript. This is an incredible loss of time for the reviewer (and editor) and very frustrating.
Response:
We thank your valuable suggestions and apologize for the upset. We will add the new line numbers where the text has been changed in the revised manuscript.
- This point is a minor issue which this reviewer had overlooked previously. I apologize. Please state in the Materials and Methods section at which time glycemia was measured concerning figure 5A. Are these fed animals? Measured at the very beginning of the light cycle? End of the light cycle? Had these animals been starved before measuring glycemia?
Response:
Thank you for your detailed suggestions. The blood was collected at 9:00 am after 12 hours of diet prohibition (from the 9:00 pm of the night).
- The mice were starved for 12 hours before measurement of the glycemia.
- Measuring was started at the beginning of the light cycle (9:00 am)
The related content has been added to the Materials and Methods section of the revised manuscript (Line 353-354).
- Same for the new figure 6. This reviewer assumes that these animals were probably starved before treatment, but this is stated nowhere.
Response:
All mice were starved for 12 hours before treatment with sucrose and corresponding extracts. The related content has been added to the Materials and Methods section of the revised manuscript (Line 360-361).
- Lines 284-285. It is stated that "compounds 11, 31, and 46 were present in roots and twigs". Then the authors conclude that it is mostly the alpha-glucosidase activity that is the primary mechanism explaining the hypoglycemic effect of these two extracts. This reviewer does not understand this reasoning. First of all, in contrast to what the authors state, the 3 compounds mentioned above are NOT PRESENT in twigs. This is a terrible mistake (and I don't think it is the job of a reviewer to correctly interpret the results of a manuscript). Please correct.
Response:
Compounds 11, 31, and 46 were present in roots and twigs (Table 1), however, the level of 11 and 46 was low and did not form the obvious peaks in the TIC of twigs. The peak number of 11 and 46 were added to the TIC of twigs (In the revised Figure 2).
These sentence has been changed to “Therefore, inhibiting the intestinal α-glucosidase activity and thereby reducing the postprandial glycemia could be a primary mechanism of hypoglycemic activity for the M. alba roots and twigs, especially for the extract of roots.” based on your suggestions. (Line 287-289)
- Related to issue 4. The resulting conclusion of this error (under 4) is therefore incorrect. Conclusion should be tuned down and be more objective.
I suggest "...inhibiting the intestinal alpha-glucosidase activity and thereby .... could POSSIBLY be a major mechanism responsible for the hypoglycemic activity of M. alba, especially in root extracts. Nevertheless, other mechanisms, for instance those involving an insulin sensitising, insulin mimicking, or insulinotropic action cannot be excluded."
If (also) a GTT had been performed as suggested by this reviewer (previous minor point #2), instead of the sucrose administration, these 3 latter possibilities could possibly have been discarded. Now, these 3 possibilities cannot be ignored.
Response:
The conclusion has been revised: “Therefore, inhibiting the intestinal α-glucosidase activity and thereby reducing the postprandial glycemia could be a primary mechanism of hypoglycemic activity for the M. alba roots and twigs, especially for the extract of roots.” based on your suggestions. (Line 287-289)
In previous studies, the hypoglycemic activity for the M. alba roots, twigs, or leaves extracts were investigated. Four main mechanisms were proposed. We think inhibiting the intestinal α-glucosidase activity and thereby reducing the postprandial glycemia could be a major mechanism based on the postprandial glycemia experiment and the previous studies. However, the three latter mechanisms may be involved in reducing the blood glucose level.
- The legend of Fig 6 lacks information. Was sucrose co-administrated with the extracts? (sucrose is not even mentioned in the legend). Orally? Both at time zero?
Response:
The sucrose solution was primarily given to the normal mice by intragastric administration, and then the extracts were immediately given using the same mean of administration. The blood in the tail was collected after intragastric administration of extracts solution, and that time was defined as zero time.
The corresponding information has been added to Figure 6 and the Materials and Methods section. (Line 292 and 363-365)
- This concerns previous minor issue #24. Apparently, as confirmed by the authors, ten mice with blood glucose levels below 11.6 mM were discarded. This should be stated in line 345, after ".....after the high-fat diet feeding."'
Response:
These sentence has been changed to “and 70 mice (Blood glucose level > 11.6 mmol/L) [34] were successfully induced to the T2D mouse model by intraperitoneal injection of STZ for three days (80 mg/kg) after the high-fat diet feeding. (Line 344-346)
- This concerns previous minor point #22. As the authors now confirm, the NC mice received normal diet (chow). This should be stated in the text (line 342), because possibly, a control group could have been given the high-fat diet without the STZ, because this would also not lead to hyperglycemia. It's just another type of control.
Response:
The NC mice fed a normal diet was added to the revised manuscript. (Line 343)
- Lines 350-351. During 28 days, mice were treated daily with extracts or controls. During these 28 days, high-fat diet was maintained? If this was the case, please state so.
Response:
During 28 days, the high-fat diet was maintained with the extracts or controls. The sentence has been changed as“Mice were treated daily with the corresponding solution and the high-fat diet for the follow of 28 days.”(Line 352-353)
- Paragraph 3.6. NC mice received water? If this was the case, please state accordingly. Were extracts/controls co-administrated with sucrose? Intragastrically? Please note these details in the text of this paragraph.
Response:
The NC mice received distilled water by intragastric administration after giving the sucrose solution. The extracts or controls was given to mice of other groups (except for NC group) immediately after supple of the sucrose solution by intragastric administration.
The corresponding details have been added to the revised manuscript. (Line 361, 363-365)
